
# Impacts of a large extra-tropical cyclonic system in
# Southern Brazilian Continental Shelf using the COAWST
# model
Luis Felipe F. Mendonça[1,6,7], Antônio F. H. Fetter-Filho[2], Mauro M. Andrade[3], Fabricio S. C. Oliveira[4],
Douglas S. Lindemann[5,8], Rose Ane P. Freitas[5], Carlos. A. D. Lentini[1,6,7].
[1] Federal University of Bahia – UFBA, Salvador, Brazil.
[2] Federal University of Santa Catarina - UFSC, Florianópolis, Brazil.
[3] University of Itajaí - UNIVALI – Itajaí, Brazil.
[4] Federal University of Rio Grande - FURG, Rio Grande, Brazil.
[5] Federal University of Pelotas – UFPEL, Pelotas, Brazil.
[6] Tropical Oceanography Group – GOAT.
[7] Postgraduate Program in Geochemistry: Petroleum and Environment (POSPETRO) – UFBA, Salvador, Brazil
[8] Postgraduate Program in Meteorology – UFPEL, Pelotas, Brazil.
*Corresponding author*: Luis Felipe Mendonça (luis.mendonca@ufba.br)
**Abstract.** The Southern Brazilian Continental Shelf (SBCS) is an area with great ecological and economic
importance to Brazil. In this region can be observed the recurrent passage of frontal systems and
extratropical cyclones, which are more frequent during the winter months of the southern hemisphere. These
systems act on the ocean surface layers as direct driving forces, which may change the thermohaline
structure of the water column and induce sea level perturbations. This study used the coupled ocean-
atmosphere regional model (COAWST) to evaluate the effect of the passage of a frontal system associated
with an extra-tropical cyclone. The ocean and atmosphere models (ROMS and WRF) was configured with
two nested grids, in order to solve the dynamic processes, at different scales, that comprise the energy
transfer from the atmospheric system to the ocean. The simulation was based on a study case, occurred in
September 2016, on the southwestern brazilian continental shelf. The model outputs were
analyzed/compared to remote sensing data and 5 tide gauges from the Agricultural Research and Rural
Extension (EPAGRI) in Santa Catarina state, Brazil. This comparison showed a correlation higher than
78% between sea level rise data and the model results, with average difference of less than 25cm. The use
of low-pass Lanczos-Cosine filter made it possible to identify the meteorological component in the ocean
model outputs. Our simulation also presents the sea level anomalies, associated with the crossing of the
atmospheric frontal system, progressing to northward along the continental shelf at 480km.day$^{-1}$, probably
associated with the presence of a coastal-trapped wave.
## 1. Introduction
The Southern Brazilian Continental Shelf (SBCS) is located between the latitudes of 25ºS and 33,74°S, it
is characterized by the frequent passage, formation, or intensification of frontal systems predominantly
from Chile, Argentina, and Uruguay (Gan & Rao 1991, Hoskins & Hodges 2005, Reboita *et al*., 2010;
Krüger *et al*., 2012, Escobar *et al*., 2016 and Gramcianinov *et al*., 2019). Several studies have shown that
the Southwest Atlantic Ocean has favorable atmospheric and oceanic conditions for cyclogenesis. These
conditions contribute to an intense exchange of sensible and latent heat between the two systems. These
phenomena have been the subject of several studies using observed data, reanalysis, global and regional



models (Bitencourt *et al.*, 2011; Gozzo e Rocha, 2013; Rosa *et al.*, 2013; Gozzo *et al.*, 2014; Rocha *et al.*,
2018; Reboita *et al.*, 2019; Reboita *et al.*, 2020).
Cyclogeneses are recurring features on the Southwest Atlantic Ocean at mid-latitudes and can be observed
throughout the entire year, being more frequent between May and September (Escobar *et al.*, 2016). The
passage of frontal systems may be accompanied by cyclones and anticyclones, which change the pressure,
wind, and other atmospheric variables. Using data from ERA-Interim, Machado *et al.* (2020) indicated that
storms on the southern Brazilian coast are accompanied by strong winds causing intense disturbance on the
sea surface. These synoptic systems have a highly destructive potential, with strong winds hitting the
coastline, lasting from hours up to days. One of the most well documented phenomenon of this kind was
the Catarina Cyclone. This system generated winds in excess of 150km.h$^{-1}$ and inflicted considerable
damage to the infrastructure of several cities along the southeastern coast of Brazil, during March 2004.
The strong winds associated with the passage of these systems have a direct impact on the sea surface and
generate instability in the mixing layer, changing the circulation and transport in the coast (Stech &
Lorenzzetti, 1992; Castro & Lee, 1995; Innocentini & Caetano Neto, 1996; Rocha *et al.* 2004).
The winds on the SBCS are a first order forcing to the local oceanic circulation, in which the wind stress
and Ekman transport are responsible for water movements at different time scales (Moller, *et al.* 2008). The
local wind accounts for 90% of sea level changes along the coast and one of this sea level rise is associated
with the meteorological tide (technically known as non-astronomical component). In the time scales of
interest in this work, the atmosphere can induce sea level variations through two mechanical effects: (i)
pressure (inverse barometer effect) of the atmosphere over the ocean (Wunsch e Stammer, 1997) and (ii)
drag (tangential tension) caused by the wind tension on the sea surface (Dean & Dalrymple, 1991).
Large variations in the coastal sea level are associated with these meteorological events that induce the
generation of coastal-trapped waves (e.g., Kelvin waves) and intensify the coastal currents system. Frandry
*et al.* (1984) analyzed sea level data and found evidence that the observed storm surges were associated
with the propagation of coastal-trapped waves, although their theoretical analysis was for a flat-bottomed
ocean, hence allowing only the propagation of Kelvin waves. Tang & Grimshaw (1995) analyzed coastal
trapped waves generated by intense atmospheric systems, such as extratropic cyclones, and their results
show that wave fields are dominated by lower-mode shelf waves. They observed that the large-scale
response is predominantly due to shelf waves, while Kelvin waves are confined to transient wave fronts.
There are not many studies investigating the formation of coastal-trapped waves in the SBCS, Saraceno *et*
*al.* (2005) showed intra-seasonal peaks in the frequencies of SST and chlorophyll, suggesting coastal
trapped waves as a possible mechanism leading to the observed variability.
Hoskins & Rodges (2005) analyzed systems that lasted longer than 2 days and moved more than 1000km
and concluded that the storm tracks in Southern Hemisphere (SH) are important for latitudinal transports
and the dynamic of the Southern Ocean. This hypothesis was derived from many studies of the SH storm
tracks that showed the genesis of eastward-moving cyclonic systems at latitudes higher than 25°S,


traversing the southwestern flanks of the subtropical anticyclones and changing the atmospheric layer over
the SBCS. In the years 2008 and 2010, the occurrence of these events caused severe damage to the facilities
and operations of the Port of Itajaí and Navegantes due to floods (Casagrande, *et al.,* 2017). The recurrence
of such phenomena requires the development of monitoring programs and tools for projecting and
monitoring the impact that extreme events can have on coastal communities.
According to Chelton *et al*. (2004) data from remote sensing of SST and wind on the surface revealed that
the ocean-atmosphere coupling mechanisms are fundamentally different in the small and mesoscale analysis
compared to the basin scales processes. Maloney & Chelton (2006), examined the ability of climate models
to simulate the positive correlation between SST and the magnitude of the wind stress at the ocean surface.
However, due to the lower resolution of the global models it is not possible to simulate the exchanges and
variations of SST and SSH at the scale of the processes necessary for a study on the continental shelf
(Taguchi *et al*., 2012). Gronholz *et al*. (2017) showed that these differences can cause notable changes to
ecosystem modeling, since the coupled ocean-atmosphere model can better determine coastal circulation,
the development of biological processes after a storm or simulate sedimentary transport over a continental
shelf. In addition, the atmospheric circulation components of the coupled model are able to simulate local
circulation patterns and exchange processes much more effectively (Cocke & Larow, 2000).
Based on this, we use a coupled ocean-atmosphere model to simulate the occurrence of a strong
extratropical cyclone formed over the southwest Atlantic Ocean, near the mouth of La Plata River, from 12
to 15 September 2016. This cyclone induced a large accumulation of water on the coast, initially in the La
Plata River next to the Uruguayan coast, transferring its energy across the SBCS like a coastal-trapped
wave. In this event, the cities of Florianópolis and Itajaí, in the State of Santa Catarina, had several areas of
flooding, as shown in Figure 1. The years 2015 and 2016 were characterized by the occurrence of the
positive phase of the El Niño South Oscillation (ENSO) phenomenon, with strong intensity.  According to
Ambrizzi *et al*. (2004) there is no evidence of an increase in cyclogenesis in ENSO years, but there is an
increase in the region where the cyclone occurs between 30°-60°S, which comprises our study area. The
southern states of Brazilian coast are prone to the frequent passage of frontal systems and the formation of
extratropical cyclones. These systems often cause flooding and socio-economic losses to the coastal areas
as described by Machado *et al.* (2011); Evans & Braun (2012) and Guimarães *et al.,* (2014).

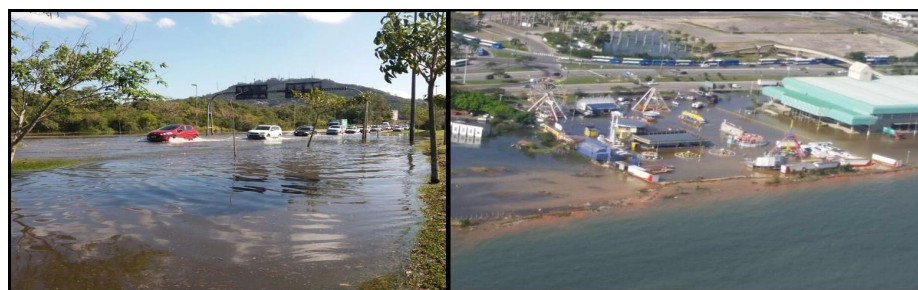

Figure 1 - Flood in the city of Florianopolis on 15 September 2016. Source: Executive Secretariat of
Communication and Civil Police Air Service.





The present study uses a regional coupled ocean-atmosphere numerical simulation, aiming to reproduce the
coastal ocean circulation during the meteorological event of September 2016. For this simulation, we used
an ocean-atmosphere coupled model with a nested grid that comprises the coast of Santa Catarina State
(Fig. 2), in Brazil. This region was chosen due to existence of tide-gauge data from the Agricultural
Research and Rural Extension (EPAGRI), kindly made available to use in the validation of the model. The
simulation results were used to investigate the impacts of energy transfer from the atmosphere to the SBCS
waters, as well as the sea level variability produced along the coastal region, during the passage of the
frontal system.
**2. Materials and Methods**
**2.1.       Characterization of the study area**
The circulation over the South Atlantic Ocean is controlled by a high-pressure center associated with the
Hadley cell, called the South Atlantic Subtropical High (SASH) (Sun, *et al.*, 2017). This system varies
seasonally and directly influences the climate in the southern and southeastern regions of Brazil, showing
a connection between SST anomalies and precipitation on the Southwest Atlantic Ocean (SWAO) (Diaz *et*
*al*., 1998). The seasonal profile of winds on the SWAO is strongly associated with the SASH latitudinal
variation. According to Hoskins & Hodge (2005), the SWAO is one of the main regions influenced by
extra-tropical cyclones, anticyclones, and droughts in the SH, especially during winter. The Southwest
Atlantic Ocean are constantly affected by the passage of cold-air masses and extra-tropical cyclones. The
ocean-atmosphere coupling in mid-latitudes under a strong wind regime induces turbulent mixing and
changes the conditions of the mass transport of marine currents over the entire SBCS (Camargo *et al*.,

128   2013).

The SBCS is characterized by a regular morphology, that presents a soft slope and establishes a smooth
passage to the upper continental slope (Zenbruscki *et al*., 1972). The ocean circulation on the SBCS is
characterized by the presence of the Brazil Current, western boundary current of the subtropical gyre of the
South Atlantic Ocean (Matano *et al.,* 2014). The Brazilian Current (BC) is formed at approximately 16°S
and flows to the south/southwest (Wienders *et al*., 2000). In the opposite direction to the BC, the Malvinas
Current (MC) is a faster current that flows to north/northeast parallel to the Argentine shelf break and carries
between 40Sv and 70Sv (Fetter & Matano, 2008; Matano *et al*., 2010). The collision of these two currents
occurs at approximately 38°S, the so-called Brazil-Malvinas Confluence (Fetter & Matano, 2008; Matano
*et al*., 2010). The shelf circulation consists of a flow of cold waters from the south – transporting the
subantarctic Shelf Water (SASW) – and a flow of hot waters from the north – carrying the Subtropical Shelf
Water (STSW) – that are constantly fed by the BC (Piola *et al*., 2000; Palma *et al*., 2008). Shelf subantarctic
waters have thermohaline characteristics that are different from those transported by the MC and their
velocities are substantially smaller (Piola *et al*., 2010).
In the SBCS the internal circulation is regulated by the local wind, while the BC plays a major role in the
mid and outer shelf regions (Palma & Matano, 2009; Matano *et al*., 2010). During the winter, winds from
the south quadrant induce mixing between the SASW and the La Plata River Plume (PRP), forming a




strongly seasonal marine current, called the Brazilian Coastal Current (BCC), over the continental shelf
region north of the La Plata River mouth (Souza & Robinson, 2004). This current is forced by the local
wind and by the river discharge from the La Plata River (average runoff of 24,000m³.s⁻¹) and Patos-Mirim
(average runoff of 2,000m³.s⁻¹) water systems, creating a unique thermohaline profile on the Uruguayan
continental shelf and on the northern SBCS (Matano *et al.,* 2014).
The astronomical tides on the SBCS have mean amplitudes of 0.4m and 1.2m for the neap and spring tide
periods, respectively. The astronomical tide is mixed with a semi-diurnal predominance and presents height
inequalities, the small tidal amplitude is associated with the proximity to an amphidromic point (for M2) in
the South Atlantic Ocean. Due to meteorological forcing, sea level may raise up to 1m above the predicted
astronomical tide (Truccolo *et al*., 2004). The southern Brazilian region is constantly affected by the
crossing, formation and intensification of atmospheric fronts. These transient meteorological systems move
over this region, changing the fields of temperature, pressure, wind, waves and are responsible for the
transfer of momentum for the formation of meteorological tides (Wallace & Hobbs, 1977). Meteorological
tides are very frequent and have different amplitudes on the southern Brazilian coast (Andrade *et al*., 2018).
Explosive cyclogenesis near the south coast of Brazil are the main causes of storm surges on the coast.
These phenomena in the region occur throughout the year, but more frequently during winter months. The
main mechanism responsible for transferring energy from the atmosphere to the ocean is through the friction
of the winds on the sea surface, which associated with the force of Coriolis and the oceanic mesoscale
gyration, cause the accumulation of water in the coast (Pugh, 1987). Storm surges amplify the occurrence
of erosive processes and floods over the coastal region (Parise *et al*., 2009). Natural disasters and extreme
events are already a reality in Brazil and the diagnosis of their occurrences are an important tool to guide
and improve policies of coastal management.
**2.2. Numerical Simulation**
The southern Brazilian coast is characterized by synoptic winds, dominated by mesoscale processes
associated mainly with the passage of frontal systems. Despite of the acceptable performance of some global
models to simulate these conditions, the oceanic characteristics of SBCS require a regional model capable
of representing the intensification of winds by processes of exchange between the ocean and the atmosphere
(Perlin *et al.,* 2011). Gronholz *et al.* (2017) showed that the interactions in the mixing layer, during and
after the crossing of a synoptic system, in an uncoupled simulation, with lower spatial resolution, do not
represent satisfactorily the shelf environments. In turn, a coupled simulation is able to mix the entire water
column with higher resolution of data. Therefore, in this study a coupled ocean-atmosphere model system
was used, in order to better represent the complex shelf circulation dynamics associated with the passage
of frontal systems.
The model setup and the validation methodology are similar to those used by Mendonça *et al.* (2017), since
the authors used the same parameterization of the ocean and atmosphere models, essential to guarantee the
quality of the simulation. The current understanding of coastal processes is largely based on numerical
models, which mathematically reproduce the dynamic conditions of a region of interest. The present study



used the coupled ocean-atmosphere modules from the *Coupled Ocean Atmosphere Wave Sediment*
*Transport* (COAWST) model system (Warner *et al*., 2008). The ocean module is composed by the *Regional*
*Ocean Modeling System* (ROMS) (Haidvogel *et al*., 2000; Shchepetkin & McWilliams, 2005, 2009) and
the atmospheric module corresponds to the *Weather Research and Forecast* (WRF Model Version 4.0.3).
WRF is a non-hydrostatic, fully compressible atmospheric model that has a range of parameterizations and
terrain-following mass-based, hybrid sigma-pressure vertical coordinate based on dry hydrostatic pressure,
with vertical grid stretching permitted (Powers *et al.,* 2017). The wave and sediment transport modules
were not used in these simulations because this paper is focused only on the oceanic component of
COAWST. Even though surface gravity waves are important for determining the momentum transfer from
the atmosphere to the ocean, this work does not include that process, for a series of reasons: absence of long
surface gravity waves data in the region to validate the model results; the poorly known topography at the
model resolution to properly resolve surface gravity waves; the introduction of one more layer of
parametrization with scarce data to corroborate the results. Adding too many degrees of freedom to a system
sometimes leads to apparently correct results, with the wrong physics. ROMS constitutes a three-
dimensional regional ocean model of free surface that uses the finite-difference method, solving the
Reynolds-Medium and Navier-Stokes equations with the Boussinesq and the hydrostatic approximations.
The model uses an Arakawa-C grid with a mask for coast delimitation and sigma vertical coordinates
(Shchepetkin & McWilliams, 2005).
The ROMS oceanic model was configured with two nested grids: a mid-resolution parent grid and a coastal,
high-resolution, child grid (Fig. 2). The parent grid was intended to solve the main meso and large-scale
circulation, as well as the set-up/set-down mechanisms associated with the wind on the SBCS and adjacent
ocean. On the other hand, the child grid was chosen to better resolve the coastal processes of the SBCS
region. The ocean parent grid comprised the coastal region between 20–40°S and 40–60°W, with a
horizontal resolution of 1/9°. The child grid embraced the region in 25–29.3°S by 46.3–50°W, with a
horizontal resolution of 1/27°, both grids had 32 sigma vertical levels. It should be pointed out that the child
grid was not centered on the parent grid, the reason being that: the location of the south open boundary of
the parent grid was chosen in order to include the La Plata River and its inflow, which are important for
determining the circulation of this part of the South American Continental Shelf; the north open boundary
of the parent grid was chosen to be far enough from the north open boundary of the child grid, in order to
avoid contamination of the child grid solution by inevitable reflections that occur at any open boundary.
The ROMS initial conditions are from Copernicus - GLOBAL Ocean Sea Physical Analysis and Forecasting
Products (GLOBAL_ANALYSIS_FORECAST_PHYS_001_024 - Global Ocean 1/4° Physics Analysis
and Forecast Updated Daily), available in *https://resources.marine.copernicus.eu*. At the three open
boundaries (N-S-E, Fig. 2) the solution was nudged to data from the same Copernicus Product which were
imposed every hour at a horizontal resolution of 1/4°. The bathymetry used was from the ETOPO1 - Global
Relief Model (Amante & Eakins, 2009) provided by the *National Centers for Environmental Information*
(NCEI). The tidal harmonic components (M2, S2, N2, K2, Q1, O1, P1, K1, M4, MS4, and MN4) from the
tidal model *OSU TOPEX/Poseidon Global Inverse Solution* (TPXO) (Egbert & Erofeeva, 2010), version




7.2, were applied at the open boundaries. This work shows the statistically stable state attained by this
model and its capability of representing the oceanographic characteristics of the SBCS.

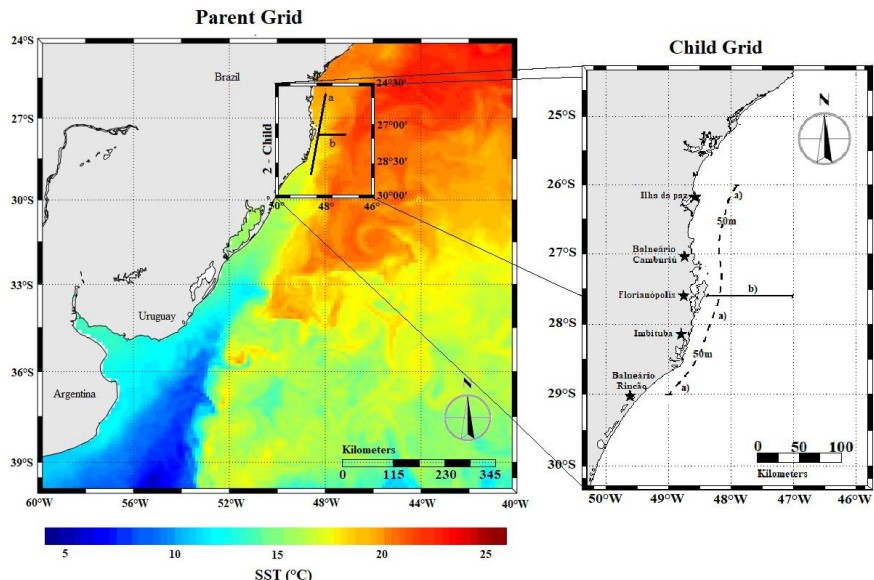


**Figure 2** – Study area and the two nested grids: a) parent and child grids and b) child grid. Colored contours
represent the mean SST modeled for the period between 11 and 19 September 2016. The child grid has 2
transects used to identify the coastal-trapped wave: a) meridional transects along the 50m isobath on the
continental shelf between 26–29°S; b) zonal transect at 27°40'S.
The WRF regional atmospheric model was configured with a similar grid to the ocean model, having a
12km horizontal resolution and 38 isobaric vertical levels. The chosen Micro Physics Option is *WRF*
*Single–moment 3–class and 5–class Schemes* (Hong *et al.*, 2004), Planetary Boundary Layer Scheme is
Yonsei University Scheme (YSU) (Hong, *et al.*, 2006), cumulus parameterization was Kain–Fritsch
Scheme (2004), shortwave and longwave parametrization is Dudhia (1989) and RRTM (Mlawer, *et al.*,
1997), respectively, Land Surface option is NOAH (Niu, *et al.,* 2011). According to Kleczek, *et al* (2014)
the adequate time for a spin-up is estimated based on the initialization conditions, that is, it can be affected
by the domain range and by the disturbances of the local limits. Based on previous sensitivity studies, the
official WRF website recommends a 12h turnaround time as the initial state. According to the developers,
this is the most suitable condition in many case studies without further verification. In this study, the model
spin-up lasted 24h and two more days of simulation before the occurrence of the extratropical cyclone
object of this study. The model was integrated for 198h with a time step of 72s and output interval of 3h –
according to the empirical calculation described by Skamarock *et al*. (2008).
To perform the coupled simulation, it is necessary to adjust the frequency with which the models exchange
information with each other via Bulk-type parameterizations. For this simulation, the interval of information
exchange between the models was 300s, so that the parameterizations and adjustments were extracted





individually from each model during the simulation. In order to stabilize the ocean model kinetic and
potential energies, one spin-up of 5 years was run as from 10 September 2011, with initial and boundary
conditions of the Mercator Global Analysis Forecast and forced by CFSv2. This process kept running until
the solution reached a quasi-equilibrium state, as suggested by Kantha & Clayson (2000). The final spin-
up conditions were used as the initial condition in oceanic model of COAWST to run the simulations from
10 to 18 September 2016, with hourly outputs subsequently validated according to the following
methodology.
**2.3.    Data validation**
To assess the hindcast accuracy of the model, two data validation steps were performed. The first one
consisted of comparing temperature and sea surface elevation from the model output with remote sensing
data. The SST data used in this comparison was from the *Optimum Interpolation Sea Surface Temperature*
(OISST), version 2 (Reynolds *et al*., 2007), which corresponds to a series of daily products of global
analysis from the *National Oceanic and Atmospheric Administration* (NOAA). The AVHRR-OI product
uses an Optimum Interpolation (OI) analysis system on the infrared data measured by the *Advanced Very*
*High-Resolution Radiometer* (AVHRR) sensor (Pathfinder versions 5.0 and 5.1) combining observations
from satellites, ships and buoys on a regular global grid. This product was available daily at a horizontal
resolution of 1/4° (Casey *et al*. 2010), spatially gridded through data interpolation and extrapolation,
resulting in a complete, smoothed SST field. Data from remote and *in situ* platforms were statistically
adjusted to fill non-existent values on satellite product, usually caused by the frequent cloud coverage in
the region, especially in winter.
The use of interpolated data and multi-channel sensors is considered fundamental to study this region. The
altimetric data were from the *Near-Real Time and Delayed Time* products from *AVISO at* 1/4° resolution,
available in *https://www.aviso.altimetry.fr*. These data consist of series of daily products of global analysis
from the *Center National d'Études Spatiales* (CNES) and is calculated with an ideal computing time
window centered at 6 weeks, before and after the acquisition date. The data acquisition was performed
through a sequence of ten daily images that corresponded to the period from 10 to 20 September 2016. The
data were stored without pre-processing and resized to meet the same number of grid points from the ocean
model in order to subsequently perform a pixel-by-pixel comparative analysis. Finally, the Bias and Root
Mean Square Error (RMSE) statistics were performed with later plotting of the observed values.
The second validation step consisted of comparing the model surface elevation (zeta) to tide gauges from
the pilotage of the Rio Grande Port, located in Rio Grande (32.13ºS–52.10ºW), pilotage of the Tramandaí
(RS), located in Tramandaí (29.98ºS–50.13ºW), Agricultural Research and Rural Extension of Santa
Catarina (EPAGRI), located in the cities of Balneário Rincão (28.83°S–49.23ºW), Imbituba (28.13°S–
48.40°W), Florianópolis (27.59ºS–48.54ºW), Balneário Camburiú (27.00ºS–48.63ºW) and São Francisco
(26.20ºS–48.50ºW). The grid point closest to the coordinates of the tide station was chosen and the analysis
of the astronomical tide and subtidal components was performed for the period from September 10 to 18,
2016. The raw data were processed using a Lanczos-Cosine low-pass filter (Thompson, 1983), which



removes 95% of oscillations higher than 40h$^{-1}$. This process is responsible for separating the high-frequency
(tidal) from low-frequency (subtidal) components that are associated with an active meteorological system.
Based on the distinction between tidal and subtidal data, the comparison and the statistical analysis of Bias
and RMSE were performed during the period of interest.
**3. Results and Discussion**
**3.1 – Atmospheric Analysis**
The analysis of the atmospheric conditions was carried out using synoptic charts from the Center for
Weather Forecast and Climate Studies (CPTEC-INPE), from 10 to 18 September 2016, available in
*http://tempo.cptec.inpe.br/cartas.php?tipo=Superficie*. Fig. 3 shows a transient synoptic system present in
the Southwest Atlantic Ocean (SWAO), significantly changing the mean atmospheric circulation of this
region. The Figure 3 describe the track of the atmospheric system since the formation of the extratropical
cyclone over the ocean, until the displacement offshore on 15/09/2016. A low-pressure center was observed
in 12/09/2016, this system moved zonally (close to 32°S) from Argentina towards Brazil. Afterwards the
low-pressure center moved southeastwards (00h on 13/9/2016), at this point as an extratropical cyclone on
the east coast of Argentina, as indicated by the letter "a". During 13/09/2016, the cyclone intensified to the
east of the Argentinean city of Mar del Plata, the sea level pressure values ranged from 996hPa at 00h to
975hPa at 00h on 14/9/2016, with a pressure drop of 21hPa in 24h, which was classified as a kind of
explosive cyclogenesis (Dal Piva *et al*., 2011; Reale *et al*., 2019; Schossler *et al*., 2020).

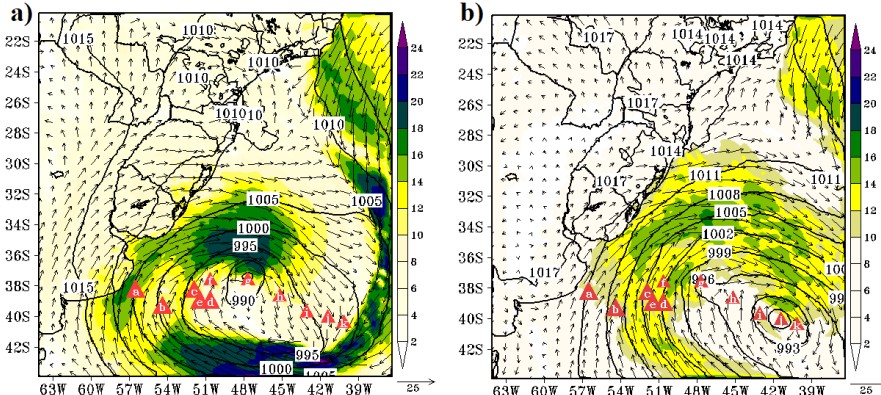


Figure 3 - Representation of the extratropical cyclone that occurred near the coast of southern Brazil,
Uruguay and Argentina, during 12-15 September 2016: a) 12h on 14/9/2016; b) 06h on 15/9/2016. Red
triangles represent the center of the cyclone every 6h, "a" represents 00h on 13/09/2016, and "k" represents
12h on 15/09/2016. Vectors indicate wind speed and direction (m.s$^{-1}$), areas filled with magnitude and black
lines indicate pressure at sea level (hPa).
Strong winds from the southwest quadrant reached the coastlines of Argentina and Uruguay during
13/09/2016 (letters "a" to "d"), reaching up to 24m.s$^{-1}$ at 18h. The intense southwest winds on 13/9/2016





piled the waters up at the mouth of La Plata River, against the Uruguayan coast. The southern part of Rio
Grande do Sul State coast (Brazil) felt the influence of the extratropical cyclone, with winds from the west
quadrant 10–12m.s$^{-1}$. The cold front associated with the extratropical cyclone caused precipitation in this
region according to meteorological stations of the National Institute of Meteorology (INMET), with
accumulations between 30–40mm. During 14/09/2016, the extratropical cyclone moved offshore and the
winds on the coast of Rio Grande do Sul began to intensify, reaching values of 14m.s$^{-1}$ at 12h, when the
cyclone was at the "f" position (Figure 3a). At 18h on 14/9/2016, when the extratropical cyclone was at the
"g" position, the southern Santa Catarina coast was hit by winds influenced by the low-pressure system,
with speeds close to 10m.s$^{-1}$. More intense winds were observed at 06h on 15/09/2016, while the cyclone
was moving offshore (letter "j"), with an average wind speed between 10–14m.s$^{-1}$ (Figure 3b), the system
lost strength during the day, reaching 10m.s$^{-1}$ at 18h on 15/9/2016, at this point the cyclone was out of the
study region.
In the afternoon of 15/09/2016, the INMET meteorological station, located in the City of Florianópolis
(A806), in Santa Catarina State (28.60°S–48,81°W), recorded winds with speeds between 10–11m.s$^{-1}$.
During the extratropical cyclone crossing, 12 to 15/09/2016, the correlations between the WRF output and
the meteorological station data were 0.57 and 0.97, for the wind speed next to the surface and sea level
pressure, respectively. Figure 4 presents the wind direction and intensity for the studied period, comparing
the data obtained through the outputs atmospheric model and the INMET automatic meteorological station
of Florianópolis (A806). The large variation in the wind direction of the meteorological station data,
required the filtering of high frequency data (Figure 4a – black arrow), so that we could better understand
the impact of the Ekman transport and storm surge. The greatest variations in sea level data and wind
intensity were analyzed in a synoptic scale for the study region. The cyclone west border remained active
in Santa Catarina coast until 16/09/2016, intensifying winds from the south-southwest (S-SW) quadrant on
the southern coast of Brazil. After 16/09/2016, the extra-tropical cyclone lost strength and moved
southeastward of the study area, maintaining its high pressure until 18/09/2016.

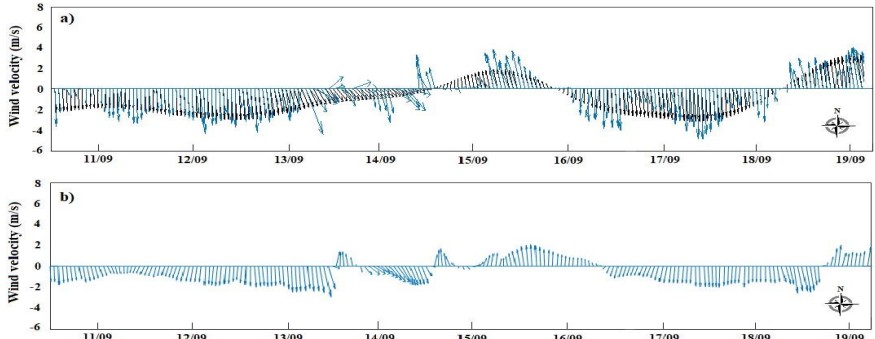


**Figure 4** – a) INMET meteorological station in the City of Florianópolis/São José (A806) - the blue arrows
indicate raw data wind speed (m.s$^{-1}$) and the black arrows are the filtered wind data, without the high
frequency of directional variations; b) output from the meteorological model (WRF) at the same position
of the INMET meteorological station.





### 3.2 Oceanic Analysis

Figure 5a shows the T/S scatter diagram, of the parent grid, with average model output values for the SBCS region, calculated from 11 to 19 September 2016. Data were plotted with the thermohaline indices described by Möller *et al*. (2008) to characterize the main water masses on the SBCS. According to this characterization, it is possible to observe that the numerical model was able to represent the main water masses in the study region: PRP, SASW, Subtropical Shelf Water (STSW), Tropical Water (TW), and South Atlantic Central Water (SACW). It should be pointed out that, even during the spring, the presence of the SASW in latitudes higher than 35°S indicates a slow retraction of the Brazilian Coastal Current (BCC, not shown) as described by Souza & Robinson (2004), Piola *et al*. (2005, 2008), Möller *et al*. (2008), and Matano *et al*. (2010). The T/S diagram data points that corresponded to the SASW and STSW were concentrated at the edges of the characterization boxes, showing that the mixing occurs gradually over the continental shelf as demonstrated by Möller *et al*. (2008) and Mendonça *et al*. (2017).

Figures 5b and 5c shows the T/S scatter diagram with the modeled data on the continental shelf of the State of Santa Catarina (child grid) before and during the passage of the atmospheric system. According to Pingree & Mardell (1981) in stratified shelf waters, the mixture caused by internal waves, due to the barotropic tides, plays a role in cooling the shelf break and in releasing nutrients. In addition to the wind induced resurgence, the increase in phytoplankton growth can occur under improved light conditions, where nutrients and phytoplankton are released into surface waters along the shelf break region. During the analyzed period, there was a large volume of Plata Plume Waters (PPW), which were transported during the winter by the BCC (Souza & Robinson, 2004). Figure 5b shows the presence of the lighter La Plata Plume Waters, with lower densities, especially at the surface. Figure 5c may be showing that more intense winds increase the mixing layer and the surface salinity, through mixing with STSW. The statistical analysis of the grid points showed that there was an increase of 8% in the total volume of STSW on continental shelf of the child grid, compared to the period before the cyclone. The deepening of the mixing layer moves the SACW towards the bottom towards the shelf break (Piola, 2008). This process shows that the increase in turbulence in the shallow shelf waters can be an important mechanism for adding nutrients to the water column, with direct impact on primary productivity as described by Alcaraz, *et al*. (2002) and Saldanha-Corrêa & Gianesella (2004).

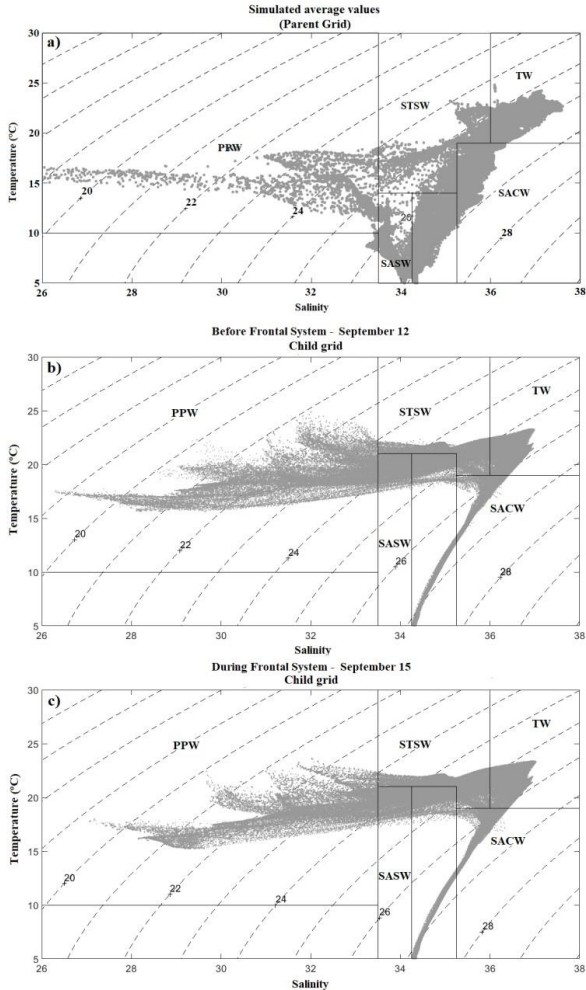

**Figure 5** – (a) T/S scatter diagram presenting the average model output data (gray dots). Shaded lines represent sigma-T values (density). The names of the water masses and the thermohaline limits were described by Möller *et al.* (2008). (b) Model output data of child grid before crossing the frontal system in 12/09/2016 and (c) after crossing the frontal system, 15/09/2016.

The differences between OISST and modeled data were relatively small, presenting a mean Bias that ranged from -1.5 to 1.5°C (Figure 6). The differences remained constant both on the SBCS and in the open ocean, showing greater values in the thermal gradient region of BMC. RMSE was between 0–1°C, which corresponds to a good agreement between model and data over the entire time series. Mendonça *et al.* (2017) obtained RMSE values higher than 2.5°C for the BMC region during 2012. The presence of eddies and meanders in the BC core, described by Mascarenhas *et al*. (1971), follows the continental shelf contours along the entire grid. High RMSEs near the BMC demonstrates the lack of skill of the model on reproducing the mesoscale variability of the BMC, in agreement with Souza & Robinson (2004). The instabilities caused


by the thermal gradient between tropical (BC) and subantarctic (MC) waters on the surface generate distinct
mesoscale features such as meanders and eddies (Legeckis & Gordon, 1982; Lentini *et al*., 2000).

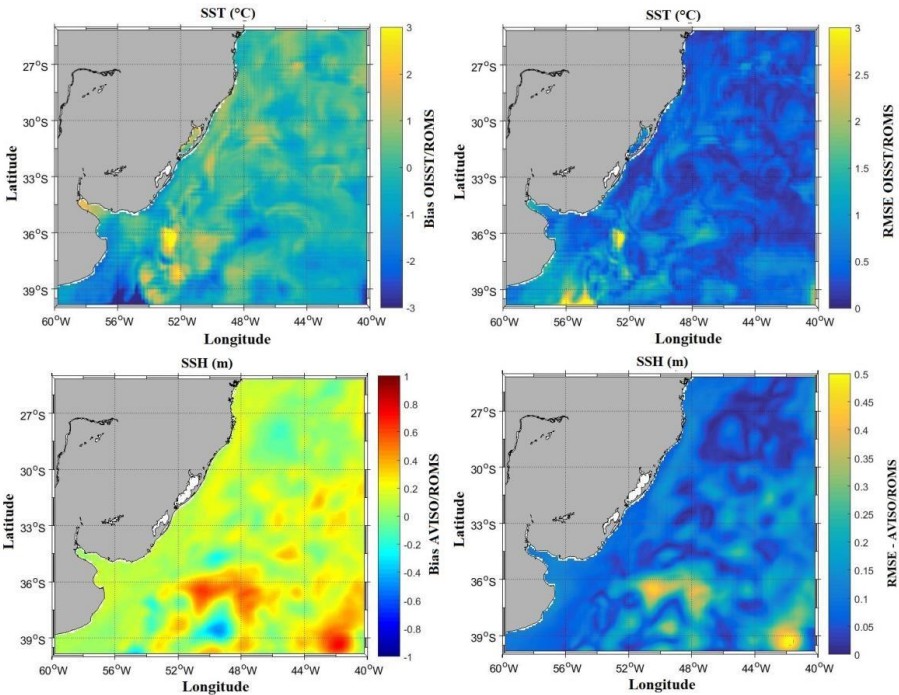

**Figure 6** – BIAS and RMSE calculated from the SST and SSH from the model outputs and remote sensing.
The comparison between satellite and modeled SSH data showed mean differences between -0.2 and 0.2m
on the SBCS and offshore, which were close to those found by Hermes & Reason (2005). The typical high
frequency periodicity in tidal data may combine with the lower sampling frequency of the altimeter and
cause distortions or associated errors, which can lead to the presence of low-frequency artificial signals in
the sampled time series (Strub *et al*., 2015). This concern with tidal errors is greater where tides are large
(Palma *et al*., 2004, 2008), although it is possible to occur significant errors in conditions similar to the area
of this study. Due to the short period of this analysis, we do not have consistent data for a more detailed
analysis of the greatest differences in sea level variation of SBCS, however the largest bias were found in
the south of the grid near the BMC, associated with the presence of meanders and large-scale eddies (Olson
*et al*., 1988).

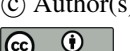



**3.3 Tide-gauge Analysis**
The second validation step of the coupled model was to compare the surface elevation values from the
model, with the data from EPAGRI tide stations in cities of Balneário Rincão, Imbituba, Florianópolis,
Balneário Camburiú e São Francisco, during the period of formation and crossing of frontal system over
the studied period. Figure 7 shows the results of the low-frequency components of all datasets, which were
obtained by filtering the data with a 39h (the inertial period of the location) low-pass Lanczos filter. The
high-frequency oscillations were obtained by subtracting the subtidal frequencies from the raw series. In
this case, there is a correlation between coastal winds and sea level, with significant coherence at all
frequencies. By analyzing the low frequency components of Figure 7, in Florianópolis, we can identify that
there is a relationship between the variation of the level and the resultant of the wind presented in Figure
4b. We observed, in the Florianópolis region, the occurrence of a synchrony between the crossing of
southern coastal winds and the rise in sea level around the 15th of September. As expected, there is a ~6h
delay between the start of the wind action and the change in sea level.
The results showed a satisfactory agreement between the time series and the model, both at the tidal and
subtidal frequency range. The largest differences appeared at high frequencies, where the model intensified
the effects of the astronomical tides. However, the model was able to simulate with reasonable efficiency
the mean low-frequency oscillations (Pearson correlation = 0,78), which are mainly forced by the
alongshore wind. Low-frequency sea level records have been widely used to study continental shelf
dynamics and its relationship with the wind and atmospheric pressure. The majority of sea level variations
associated with low-frequency subtidal components are driven by wind stress and Ekman transport
(Truccolo *et al.*, 2004). These oscillations cause a cross-shelf barotropic gradient, which is geostrophically
balanced with the alongshore current (Thompson, 1981; Stech & Lorenzzetti, 1992).




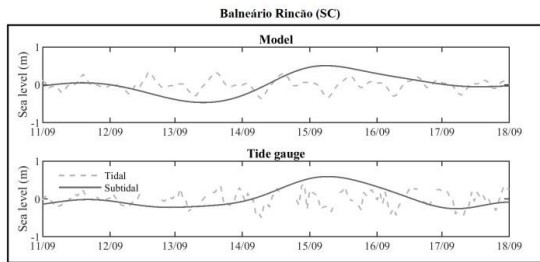

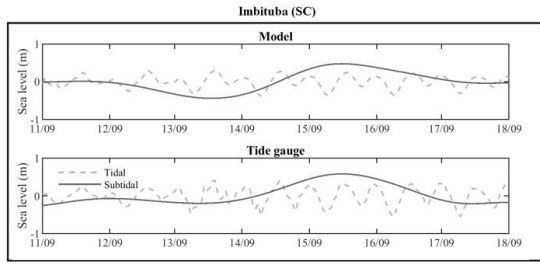

**Figure 7** - Temporal variation of the tidal and subtidal components of sea level from the numerical model and from the tide gauges of EPAGRI in the cities of Balneário Rincão, Imbituba, Florianópolis, Balneário Camburiú e São Francisco.

### 3.4 Coastal Wave Occurrence

Tang & Grimshaw (1995) showed that transient atmospheric systems can induce the accumulation of coastal waters, which are formed through the geostrophic flow generated by coastal trapped waves. In our work, the analysis of the extratropical cyclone indicates that the atmospheric system generated a sea level variation by two mechanical effects: (i) atmospheric pressure on the ocean (normal tension) and (ii) drag caused by the wind tension on the sea surface (tangential stress). The process of sea level rise started in the La Plata estuary, after the fast formation of the low-pressure core of the extratropical cyclone, as shown in Figure 3. The atmospheric pressure drops, quickly, from 1008hPa (12/09/2016 – 12h) to 990hPa 24h later. According to Pontes & Gaspar (1999), the response of sea level to surface atmospheric pressure is commonly followed by an inverted

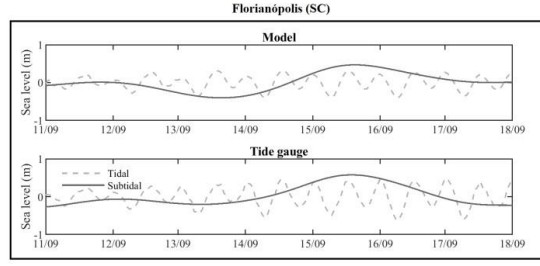

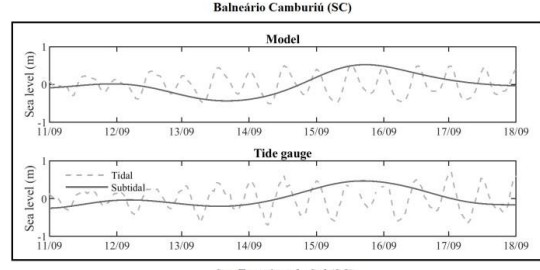

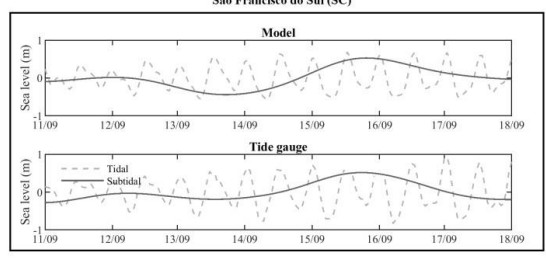

barometer phenomenon, with an estimated elevation of 1cm for each 1hPa, indicating an induced local
elevation of ~0.2m. Figure 8 shows that at 18h, on 13/09/2016, the elevation in La Plata River mouth was



approximately 0.5m, near the Uruguayan coast. Therefore, in addition to atmospheric pressure, the
horizontal movement of the south wind caused by the western boundary of the cyclone, disturbed the
equilibrium condition of the ocean. The stress transmitted into the fluid promoted a momentum transfer
from the atmosphere to the ocean (Dean & Dalrymple, 1991), overcoming the inertial period for the region
of ~20h, increasing the effect of wind set-up on the coast.
The physical mechanism that explains the force of the coastal-trapped waves over topography is
straightforward, wind stress along the coast causes transport within the Ekman layer. Then mass
conservation generates a compensating offshore flow in the lower layer. As this flow crosses the isobaths,
it leads to changes in the local relative vorticity, which is expressed in terms of velocity along the coast and,
in the presence of variability along the coast, it generates the propagation of the waves (Brink, 1991). In the
southern hemisphere, topographic Rossby waves propagate with the coast at its left, thus progressing to the
north along the western continental margin of South America. Gill & Clarke (1974), Schumann (1983), and
Batisti & Hickey (1984) argued that the alongshore component of the wind stress is the main creator and
enhancer of such trapped waves. Consequently, those waves share the same periods with the meteorological
systems that caused them. The winds formed by the gradient of the low-pressure centers created by the
cyclone are the main modulating mechanisms of positive variations in sea level (Calliari *et al*., 1998). At
00h, on 14/09/2016, it was possible to observe the displacement of a surface elevation anomaly, originated
at the Uruguayan coast, towards north (Fig. 8). The analyses of the surface elevation anomalies shows that
the winds of the extra-tropical cyclone intensified the coastal trapped wave, which continued to propagate
towards the State of Santa Catarina even after the winds ceased (15/09/2016 -18h).

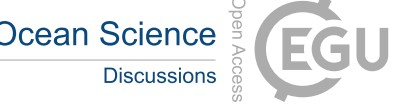

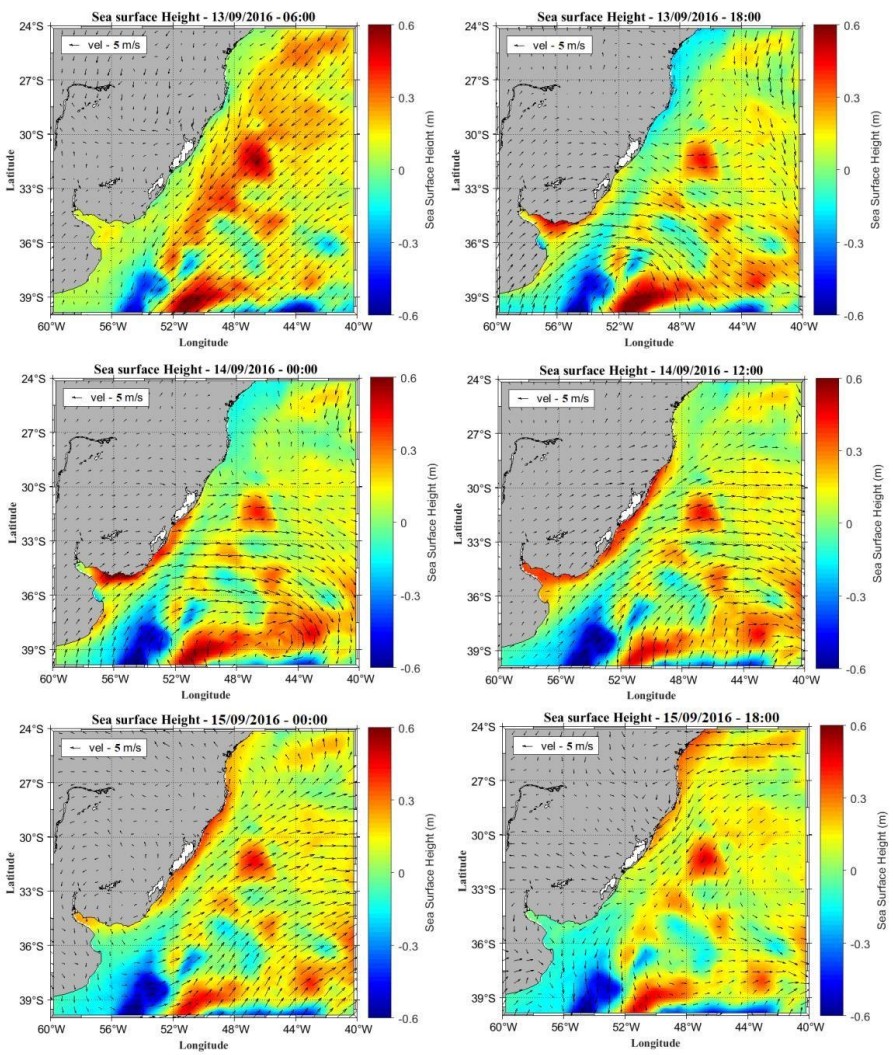

Figure 8 - Maps of the subtidal components of sea surface elevation and wind fields generated by the
numerical model, at specific dates, during the passage of the frontal system.
Melo *et al*. (2008) showed that extra-tropical cyclones formed far from the Brazilian coast can reach it and
induce significant changes in sea level and Parise (2009) identified an extra-tropical cyclone in the South
Atlantic Ocean that contributed over 70% of the energy of the wave spectrum along the coast of RS. In order
to show the propagation of the costal-trapped wave, the subtidal components from the EPAGRI tide gauges
(Figure 9a) and from the numerical model (Figure 9b) were plotted over time. As EPAGRI tide stations
cover only the Santa Catarina state, we can see in Figures 3 and 8, that the probable formation area of this
trapped wave over the River Plate, near the Uruguayan coast. Thus, were also plotted for numerical model,





in dashed lines, data on the coast of Montevideo (UY), Rio Grande (BR) and Paranaguá (BR) for analysis
of wave propagation before and after the passage through the State of Santa Catarina.
Figure 9 shows the SSH anomalies associated with the storm surge where it is possible to identify four peaks.
They present a time lag of a few hours and an advance towards the north, probably associated with a coastal
trapped wave. A cross-correlation analysis made it possible to identify the time lag of these elevations along
the coast. Through the average time of signal movement between analyzed points it is possible to estimate
the average speed of propagation in relation to the distance covered by the wave. Using the distance among
the sites and considering that the peak of maximum correlation indicates the wave time lag, it was possible
to estimate the average propagation speed of the trapped wave generated by the storm surge. Therefore,
since the cyclone formed near the La Plata river mouth, this site was considered as the region where the
winds originated or intensified the amplitude of the analyzed trapped wave. The numerical model generated
a coefficient of determination in the range of 78%, slope of the regression line close to the ideal value (1.00),
with a standard deviation of less than 10 cm. According to Brink (1991), the existing models tend to
underestimate the amplitude of observed fluctuations in the current along the coast and at sea level, usually
by around 10-50%. The wave propagation velocity analyzes showed higher velocity values (~ 10.6 m.s$^{-1}$)
over the continental shelf of Uruguay and Rio Grande do Sul. Keeping more than 10 m.s$^{-1}$ until Imbituba.
From Imbituba, until Paranaguá, the speed values decreased to ~6 m.s$^{-1}$ due, we believe, to the change in
the direction of the coastline, reduction of the slope and the widening of the continental shelf.
The integration was carried out over the eight locations represented in Figure 9. The speed of the phase
decreases over time with the reduction of winds from the extratropical cyclone. We can see that the sea level
rise in Montevideo (UY) starts around 12h on 12/09/2016, possibly forced by the effect of the inverted
barometer observed in the pressure fields of the WRF model. The advance of the low frequency coastal
wave is seen in Figure 8, in tide gauge and the model outputs, following the wind fields (S-SW) of the extra-
tropical cyclone and seems to be responsible for the formation and/or intensification of these waves on
SBCS. In the comparative analysis between the tide gauges data and numerical model we can observe similar
periods of maximum amplitudes. However, in the cases of Balneário Rincão and Balneário Camburiú, there
were overestimations of the model up to 0.1m above that observed by the tide gauge. We believe that these
differences may be associated with the position of the coastline or the collection of data in more external
areas in relation to the original position of the tide gauge. Based on the signal speed along the coast, we can
see that there are similarities in the energy distribution between each of the stations observed, through a
coherence and phase relationship along the continental shelf of Santa Catarina.
Figure 9 shows a good qualitative agreement between the data and the model, there are, however, some
important differences between model and observations. The data and the model show an evident propagation
of the SSH signal towards north, with the wave first reaching the southern cities of the study region. The
model, however, propagates the wave faster than it was observed from the tide gauges data, it is clear that
the model maximum elevation leads the tide gauge data by a few hours. This is probably due to the poor
topography of the region and possibly the limited model resolution. The tide gauge data also shows a fast



reduction of the wave amplitude after passing the city of Florianópolis and arriving at Balneário Camboriú
(less than 100km to the north).

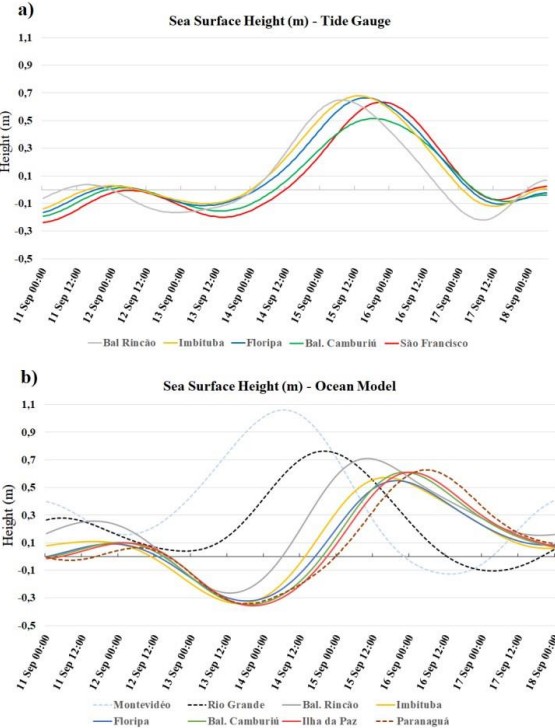

**Figure 9** – Temporal variation of SSH data: a) tide gauges in Balneário Rincão, Imbituba,
Florianópolis, Balneário Camboriú e São Francisco and b) numerical model in Montevideo, Rio Grande,
Balneário Rincão, Imbituba, Florianópolis, Balneário Camburiú, Ilha da Paz e Paranaguá, SC. Maximum
amplitude of waves for the analyzed period.
Figure 10 shows the Hovmöller diagrams of sea surface elevation, during the period from 11 to 19 September
2016. Figure 10a was generated using data from an alongshore transect (Fig. 2) over the 50m isobath –
without the tidal component – in order to evaluate the evolution of the wave trapped to the continental shelf
of Santa Catarina. The wind started changing its direction from the north-northeast (N-NE) quadrant to the
S-SW one after 00h on 13/09/2016. The sea surface elevation followed the S-SW wind fields and at 00h on
14/09/2016 the consequent impact of the storm surge was observed in the coastal waters of Balneário
Rincão. The gradual advance of this elevation to the northeast with a coherent period suggests the formation
of a coastal-trapped wave, as noted by Houghton & Beer (1976), Brink (1991), Battisti & Hickey (1984),
Yao *et al.* (1984), Kitade & Matsuyama (2000) and Junker *et al.* (2019).
The retraction of the coastline, close to 27.5°S, induced this wave to move away from the coast, but without
a significant increase in depth. The widening of the continental shelf in this region increased the elevation





area offshore, without significantly reducing the wave amplitude. The subtidal component generated SSH
values above 0.5m as seen on the Hovmöller diagram along the entire Santa Catarina coast. That agreed
with the tidal data released by the EPAGRI company from region, which recorded a residual height close to
~0.6m associated with coastal wave created by storm surge. Figure 10b shows the Hovmöller diagram of
free surface variation from a cross-shore profile near the coast of Florianópolis (28°S). The SSH values were
under a tidal regime. These data showed high-frequency oscillations and elevations of up to 1m during
maximum S-SW wind intensity. Despite the astronomical tide daily oscillations, the storm surge component
had a significant impact on sea level rise in this region. According to Truccolo *et al*. (2004), low-frequency
sea level oscillations play an important role along the coast of SC because the astronomical component is
microtidal. It is even possible to observe the augmentation of the neap tide during the period when the N-
NE winds are prevailing on the region.

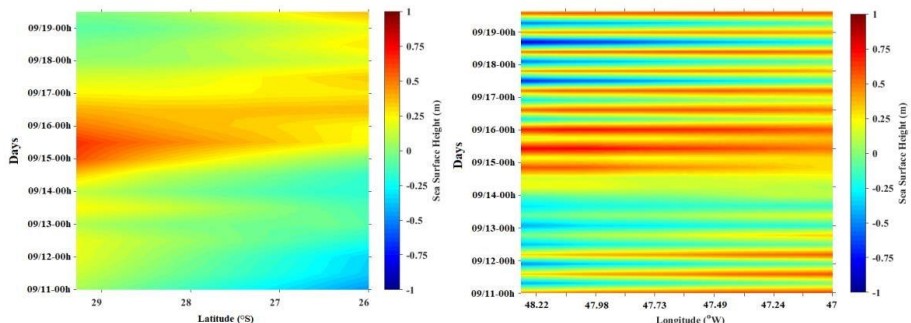


**Figure 10** – Hovmöller diagrams of SSH temporal variation. (a) Diagram generated without the tidal
components from an alongshore transect, over the 50m isobath, on the coast of Santa Catarina state, between
the latitudes of 26 and 29°S. (b) Diagram generated from a cross-shore transect over the latitude of 28°S
with the tidal components.
According to Möller *et al*. (2008) and Piola *et al*. (2008), the local wind changes the current speeds and
transport along the SBCS. Costa & Möller (2011) and Andrade *et al*. (2016) used data from acoustic profilers
to show that the direction of currents on the SBCS oppositely change throughout the water column a few
hours after the change in wind direction. Dias *et al*. (2014) added that it is not only the offshore Ekman
transport that is responsible for the physical-chemical changes on the SBCS, but the coastline geometry is
also important. Studies such as Figueiredo Jr. (1980), Zavialov *et al*. (2002), Costa & Moller (2011), and
Andrade *et al*. (2016) have shown the impact that the passage of an intense atmospheric system can cause
on the local hydrodynamics, especially on shallow waters driven by winds that are associated with the frontal
system. Rodrigues *et al*. (2004) showed that cold fronts move on the SBCS from southeast to northeast with
a monthly frequency of 3-4 fronts/month, usually followed by mobile cyclones and anticyclones (Wallace
& Hobbs, 1977). Therefore, the changes caused by these atmospheric systems have a direct influence on the
coastal environment, altering the thermohaline structure and oxygenating the coastal waters. The increase





in the number of studies on the impact of the passage of frontal systems on the SBCS may, in the near future,
provide new answers to still unresolved questions regarding the southern Brazilian coastal system.
**4. Conclusions**
The present work presented the implementation of the coupled ocean-atmosphere model for studying a
specific process of ocean-atmosphere interactions on the continental shelf from the state of Santa Catarina,
during the passage of a frontal system associated with an extra-tropical cyclone in September 2016. The
solution used boundary components adjusted for the conditions from an earlier study (Mendonça *et al.*,
2017). The presented findings demonstrated regional circulation patterns strongly associated with the
meteorological conditions present in the southern Brazilian region. These processes change the physical and
chemical properties of sea water, changing the mixing layer and fertilizing the shelf waters.
The analyzed period consisted of the rapid formation of an extra-tropical cyclone on the parent grid,
intensifying the south-quadrant winds and the associated Ekman transport. The rapid drop in pressure levels
on the La Plata River has generated an increase in sea level, caused by the inverted barometer, that was
intensified by the winds from the south-southwest quadrant. The low-frequency components were found to
be the controlling mechanism of the model elevation output. The filtering of the model altimetric data
allowed the identification of the storm surge component (low frequency) with a short lag in relation to the
tide gauge data. This process was important to identify the advance of a coastal wave (Figure 8 and 9),
intensified by the winds from the frontal system responsible for flooding several cities along the coast of
Santa Catarina.
The analysis of maximum correlation between the peaks of sea level elevation – at the eight analyzed sites
was superior to 82%. Although the relationship between wind forcing, wave, and the resulting flux is not
linear, the correlation coefficient gives us a measure of the wave propagation behavior on the continental
shelf (Fewings, 2007). According to Brink (1991), a qualitative assessment delineated the conditions for the
development of large waves trapped to the coast and associated with tropical cyclones: when the storm
moves for several hours, its translation speed slowly increases, continuously matching the group velocity of
the waves under the storm. The strong correlation between winds and the current at the surface and on the
bottom shows a rapid response of the water to wind forcing. The results obtained with the application of the
COAWST model agreed with the estimates from other authors although the dynamic arguments of the
momentum equation were not considered due to the size limitations of this manuscript. Since the southern
region of Brazil is constantly affected by the passage of frontal systems, studies capable of interpreting the
changes in the SBCS dynamics have great importance for the ecological characterization and the prevention
of coastal impacts associated with sea level variations.



**Acknowledgments**

598        We thank CNPq and INCT-Mar COI for funding in the form of a postdoctoral fellowship. The main
author, as a faculty member, would like to thank the Pos-graduate Program in Geochemistry of Petroleum
and Environment (POSPETRO) at the Federal University of Bahia. Thank all the researchers and students
of the Laboratory of Dynamic Oceans of the Federal University of Santa Catarina. To thank the Agricultural
Research and Rural Extension Company of Santa Catarina, which provided free of charge the tide data and
Dr. Elírio Ernestino Toldo Junior (Federal University of Rio Grande do Sul) for the sea level data provided
together with Prof. Dr. Mauro Michelena Andrade (Univali). The output data of the numerical model used
in this study are available in web repository (https://1drv.ms/u/s!AhHBbFlJjz9g7BapIYB3ifwgFqqSA-
?e=RqKgSh).

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
