# Peer review of "Impacts of a large extra-tropical cyclonic system in Southern Brazilian Continental Shelf using the COAWST model"

_Ocean Science, 2021_

## Community Comment (CC1)

**Title:** Impacts of a large extra-tropical cyclonic system in Southern Brazilian Continental Shelf using the COAWST model

**Construction and Building Materials**

*Dear Editor and Anonymous Referees*

*The authors thank the reviewer #1 for the suggestions regarding the submitted manuscript. All comments were carefully considered by the authors and applied to the manuscript. The following pages show the changes performed. Comments from the reviewers are presented followed by the response from the authors and the changes performed in the manuscript.*

*Sincerely,*

*The Authors*

**Revision of the manuscript:**

**This study uses the COAWST model to study the influence of an atmospheric frontal system to the coastal ocean. The frontal system is associated with an extra-tropical cyclone. The model is based on the ocean circulation model (ROMS) and the weather forecast model WRF using nested grids.**

*The use of coupled modeling systems (COAWST) to study the ocean and atmosphere physical processes at mesoscale and sub-mesoscale has previously been justified in studies by some authors. Gronholz et al. (2017), showed that exchange processes generate notable changes to ecosystem modeling, since the coupled ocean-atmosphere model can better determine coastal circulation, the development of biological processes after a storm surge or simulate sedimentary transport over a continental shelf. In addition, the atmospheric circulation components of the coupled model are able to simulate local circulation patterns and exchange processes* much more effectively (Cocke & Larow, 2000). The COAWST coupled *model has been widely used in studies of cyclogenesis and storm tracks over the ocean and continental shelf. Authors such as Olabarrieta, et al. (2012), Zambon & Warner (2014), Liu et al. (2015), Du et al. (2015), Chen et al. (2017), Denamiel et al. (2019), and Reffit et al. (2020), were fundamental for choosing this model. The icing on the cake here is the ability of COAWST to simulate energy transfers from atmosphere to the ocean on a regional grid.*

**The paper presents some model-data comparison for an event happened in Sep. 2016 and shows the area's sea level anomaly which the authors conclude that it is "probably" due to coastally-trapped waves with a propagation speed of 480 km/day northward.**

We believe that there was a misconception in writing, where the word "problably" should be replaced by "we suggest". Currently, there are few works with coastal trapped waves on the Brazilian coast, being recent and not conclusive. Due to the absence of an effective oceanographic observation network using buoys, tide gauges and drift data, we do not have access to a greater amount of in situ data. These data are essential for understanding the dynamic processes over the continental shelf and computing parameters of these special waves. Thus, we suggest that the observed and modeled free surface variations and wave propagation, associated with low frequency values of storm surge, close to observed in other

manuscripts, such as: Wang & Mooers (1977); Schumann & Brink (1990); Camayo & Campos (2006); Schlosser, et al. (2019).

**From what I read, the paper looks like a modeling practice and the results are presented without a clear discussion of the useful dynamics in detail with confidence. In fact, no quantitative analysis to the model output is done. The sea level anomaly should have been analyzed before concluding it is coastally trapped waves and if the propagation speed matches theory.**

*We appreciate the reviewer's suggestion, we added some paragraphs that collaborate with the reviewer points.*

**The figures are of low quality and captions not uniform (some are clear and the others are not).**

*We appreciate the reviewer's suggestion, and we will improve the quality of the images.*

**Some examples of problems, among many others, are this comparison showed a correlation higher than Line 28: "78% between sea level rise data and the model results," - sea level rise data? Isn't this a work for an extratropical cyclone that only lasted for a few days? Why was the short time event related to sea level rise which would be a climate data defined to be 30 years or longer. I guess the authors meant water level data.**

We understand the reviewer point, whereupon "sea level rise" is predominantly used in studies of mean sea level, associated with climatological events. We appreciate the reviewer suggestion and we changed it in the manuscript.

**Figure 1 is out of the context - it maybe useful for a conference for background but not needed as they have nothing to do with the dynamics and the coastally-trapped waves.**

*We appreciate the reviewer's suggestion and remove figure 1.*

**Figure 4, poor quality**

*We appreciate the reviewer's suggestion, and we will improve the quality of this image.*

**Figure 6, Caption is too brief and unclear.**

*We appreciate the reviewer's suggestion, and we will adjust the figure caption.*

**Figure 7, poor quality - but the presentation is odd - I would prefer to see direct comparison between model and data, not separating the tidal and non-tidal parts. If you need to separate them, put the data and model in the same frame and include quantification of statistics (e.g. correlation or R2 value).**

*The authors follow the reviewer's opinion and change the figure to a direct comparison between model and tide gauge data, not separating the tidal and non-tidal parts.*

**Lines 456 and 457: "The physical mechanism that explains the force of the coastal-trapped waves over topography is straightforward,..." then the authors basically referred to some previous papers and finishes their analysis for the coastally-trapped**

**waves. I would not call this paragraph an analysis. The work is not done with quantified analysis and any new finding in dynamics.**

*We added two paragraphs to improve the process ideas generated by low frequency waves under the influence of a storm surge. The use of literature to justify our results is due to the absence of a more in situ data and we tried to identify results observed in other authors to prove the effectiveness of our statements.*

**Similar problems exist for analysis around Figure 9. Figure 9 is presented in an odd way as well. It is not helpful in providing a clear picture.**

*The authors understand that the figure is simple, however the methodology accompanies other manuscripts, which used similar figures to visualize the propagation of the wave over time. Authors such as: Horsburgh & Wilson (2017); Brown et al. (2011); Brown, et al. (2013); Choi et al. (2013); Xie et al. (2016); Chen et al (2017); Song et al. (2020);*

**For the discussion around Figure 10: why the left panel has no tide while the right panel has tide? What is the reason to not include tide for the along shelf 50-m contour line transect but do include tide for the cross shelf transect?**

*Figure 10 shows the propagation profile of the low frequency wave, generated by the storm surge. In this way, we remove the tidal component to follow the wave propagation along the southwest coast of Brazil. Figure 10b is presented with the tide component to identify the influence of the low tides on the free surface level of the output model. This explains why the flooding on the Florianópolis island was observed in less than 24 hours. Although the elevation caused by the meteorological tide has a longer period, the M2 tide component is responsible for the semi-diurnal suppression of the relative sea level.*

**REFERENCE:**

Brown, J. M., Bolaños, R., Wolf, J. (2011). Impact assessment of advanced coupling features in a tide–surge–wave model, POLCOMS-WAM, in a shallow water application. Journal of Marine Systems, 87(1), 13-24.

Brown, J. M., Bolaños, R., & Wolf, J. (2013). The depth-varying response of coastal circulation and water levels to 2D radiation stress when applied in a coupled wave–tide–surge modelling system during an extreme storm. Coastal Engineering, 82, 102-113.

Camayo, R., & Campos, E. J. (2006). Application of wavelet transform in the study of coastal trapped waves off the west coast of South America. Geophysical research letters, 33(22).

Chen, W. B., Lin, L. Y., Jang, J. H., & Chang, C. H. (2017). Simulation of typhoon-induced storm tides and wind waves for the northeastern coast of Taiwan using a tide–surge–wave coupled model. Water, 9(7), 549.

Choi, B. H., Min, B. I., Kim, K. O., & Yuk, J. H. (2013). Wave-tide-surge coupled simulation for typhoon Maemi. China Ocean Engineering, 27(2), 141-158.

Cocke, S., & LaRow, T. E. (2000). Seasonal predictions using a regional spectral model embedded within a coupled ocean–atmosphere model. Monthly Weather Review, 128(3), 689-708.

Denamiel, C., Šepić, J., Ivanković, D., & Vilibić, I. (2019). The Adriatic Sea and Coast modelling suite: Evaluation of the meteotsunami forecast component. Ocean Modelling, 135, 71-93.

Du, J., Larsén, X. G., & Bolaños, R. (2015). A coupled atmospheric and wave modeling system for storm simulations. In Proceedings of EWEA offshore conference, Copenhagen.

Gronholz, A., Gräwe, U., Paul, A., & Schulz, M. (2017). Investigating the effects of a summer storm on the North Sea stratification using a regional coupled ocean-atmosphere model. Ocean Dynamics, 67(2), 211-235.

Horsburgh, K. J., & Wilson, C. (2007). Tide-surge interaction and its role in the distribution of surge residuals in the North Sea. Journal of Geophysical Research: Oceans, 112(C8).

Olabarrieta, M., Warner, J. C., Armstrong, B., Zambon, J. B., & He, R. (2012). Ocean–atmosphere dynamics during Hurricane Ida and Nor'Ida: An application of the coupled ocean–atmosphere–wave–sediment transport (COAWST) modeling system. Ocean Modelling, 43, 112-137.

Reffitt, M., Orescanin, M. M., Massey, C., Raubenheimer, B., Jensen, R. E., & Elgar, S. (2020). Modeling Storm Surge in a Small Tidal Two-Inlet System. Journal of Waterway, Port, Coastal, and Ocean Engineering, 146(6), 04020043.

Schumann, E. H., & Brink, K. H. (1990). Coastal-trapped waves off the coast of South Africa: generation, propagation and current structures. Journal of Physical Oceanography, 20(8), 1206-1218.

Schlosser, T. L., Jones, N. L., Musgrave, R. C., Bluteau, C. E., Ivey, G. N., & Lucas, A. J. (2019). Observations of diurnal coastal-trapped waves with a thermocline-intensified velocity field. Journal of Physical Oceanography, 49(7), 1973-1994.

Song, H., Kuang, C., Gu, J., Zou, Q., Liang, H., Sun, X., & Ma, Z. (2020). Nonlinear tide-surge-wave interaction at a shallow coast with large scale sequential harbor constructions. Estuarine, Coastal and Shelf Science, 233, 106543.

Wang, D. P., & Mooers, C. N. (1977). Long coastal-trapped waves off the west coast of the United States, summer 1973. Journal of Physical Oceanography, 7(6), 856-864.

Xie, D. M., Zou, Q. P., & Cannon, J. W. (2016). Application of SWAN+ ADCIRC to tide-surge and wave simulation in Gulf of Maine during Patriot's Day storm. Water Science and Engineering, 9(1), 33-41.

Zambon, J. B., He, R., & Warner, J. C. (2014). Investigation of hurricane Ivan using the coupled ocean–atmosphere–wave–sediment transport (COAWST) model. Ocean Dynamics, 64(11), 1535-1554.

---

## Community Comment (CC2)

**Title:** Impacts of a large extra-tropical cyclonic system in Southern Brazilian Continental Shelf using the COAWST model

**Construction and Building Materials**

*Dear Editor and Anonymous Referees*

*The authors would like to thank reviewer #2 for the suggestions regarding the submitted manuscript. All comments were carefully considered by the authors and applied. His particular corrections were fundamental for the improvement of the manuscript. Reviewer's comments are presented, followed by the response from the authors and the changes performed in the new version of the manuscript.*

*Sincerely,*

*The Authors*

**Revision of the manuscript:**

**This manuscript deals with the impact of an extra-tropical cyclonic system in the Southern Brazilian Continental Shelf using numerical modeling. Although this is an important topic of study, this manuscript presents some problems that do not allow a dynamical understanding of the problem. The main problem of the manuscript is that the numerical simulations present results that do not well represent the observations. For this reasons, mainly, I suggest the rejection of the manuscript. Below, I present in more details the problems that I found:**

**-Line 69: By shelf waves, the authors mean Continental Shelf Waves?**

*Yes, we carefully make it clear by adding Continental Shelf Waves in this sentence.*

**-Line 132: Brazilian Current or Brazil Current?**

*We appreciate the reviewer's suggestion. It now reads "The Brazil Current".*

**-Lines 150-153: There are no references regarding the tidal amplitudes.**

*References have been included: (Truccolo et al., 2004), (Pennings et al., 2012) and (Benavente, et al. 2006; Maia, et al. 2016)*

**-Line 159: How near the coast of Brazil?**

*It now reads: "*Explosive cyclogenesis over the southern Brazilian coast are the main causes of storm surges and occur throughout the year, but mainly during winter months."

**-Lines 162-163: What do the authors mean by oceanic mesoscale gyration? And why does it cause accumulation of water in the coast?**

*We adjusted the citation by Pugh (1987). Now it reads: "The main mechanism responsible for transferring energy from the atmosphere to the ocean is through the friction of the winds on the sea surface (Pugh, 1987), which associated with the Coriolis Force and Ekman transport mechanisms, can induce the water accumulation on the continental shelf."*

**-Lines 164-166: The authors could give some examples of natural disasters and extreme events in Brazil to put the manuscript in perspective.**

*It has been added (lines 101-104 and line 166) to meet the reviewer's request.*

**-Line 174-175: Models do not provide data.**

*We accept the reviewer's suggestion and remove this sentence.*

**-Lines 180-181: The authors mention the importance of models in understanding coastal processes but do not mention any of these studies.**

*We appreciate the reviewer's point. We added some examples of existing studies on this topic.*

**-Line 250 (Data validation): The authors did not validate the data in this section.**

*We change "Data validation" to "Model output data analysis".*

**-Line 408: The authors present a correlation coefficient (Pearson's) of 0.78, but no significance level. I don't believe this correlation coefficient for SSH low-frequency variability is satisfactory. Other studies dealing with similar problems present much higher correlation coefficients for the low-frequency variability (e.g., Costa et al., 2019, Khalid et al. 2020, Ruiz et al., 2021). Why not computing some other statistical parameter that deals with the comparison between actual values and not the variability alone? (e.g., Willmott, 1981)**

*The use of Pearson's coefficient was due to its ability to measure the correlation (positive or negative) in a specific predefined interval. In this way, the method proves to be effective in comparing harmonic oscillations recorded in the low-frequency component. It assists in the identification of a possible delay of the model in relation to the tide gauge. We*

*believe that correlation coefficients such as Pearson or Skill are accepted scientifically and can present satisfactory results. We appreciate the reviewer for his observation. We identified that an error occurred in the writing of the calculated correlation coefficient. The calculation was redone, and the correct value is 0.87, which is close to what has been observed by Costa et al. (2019), Khalid et al. (2020), and Ruiz et al. (2021).*

*Ruiz et al. (2021) explain in their's work: "The sub-inertial response to wind forcing is trapped in the continental shelf; sub-inertial elevations and currents variability are higher at the coast than at the shelf break (Dottori and Castro, 2009). The inclusion of the remote wind stress as forces increase the correlations between observed and modeled sub-inertial currents in the central and northern parts of the South Brazil Bight (Dottori and Castro, 2018)." In our study area, we observed that there is a change in the direction of the coastline, which induced a delay of 3 hours in the propagation of the sub-inertial wave. (Figure 6 updated). Another critical factor in reducing the correlation coefficient of the low-frequency wave is that the pre-frontal winds were more intense in the model's outputs compared to the weather station. It explains the lower levels observed in the numerical simulation.*

**-Page 15: The figures should present data and model in the same panel.**

*Figures have been modified. We gathered the data from the tide gauge model.*

**-Lines 509-511: The authors should present a figure with the locations mentioned in the text, as well as the with of the continental shelf in the domain.**

*The analyzed stations are mentioned in the text (Lines 509-511) and are shown in Figure 1 through the start icon. We appreciate the reviewer's suggestion and add the bathymetry dashed lines to mark shelf break in the child grid of Figure 1.*

**-Line 512: It is hard to see the agreement between data and model. The authors should provide a qualitative method and a better figure.**

*We understand that the figure is objective and straightforward; however, the methodology accompanies other manuscripts, which used similar figures to visualize the propagation of the wave over time. The authors understand that the more robust quantitative analysis may better represent the characteristics of the model's shelf wave. However, the absence of a greater number of observational data could assist in the comparative study of the data. Thus, we found other authors who used similar statistical analyzes such as Horsburgh & Wilson (2017); Brown et al. (2011); Brown, et al. (2013); Choi et al. (2013); Xie et al. (2016); Chen et al (2017); Song et al. (2020); Khalid et al. (2020) and Ruiz et al. (2021), explaining its results with images such as figure 8 (Updated).*

**-Line 516: The delay of few hours is considerably high for these processes. These differences can lead to substantial errors in SSH prediction that do not allow a good understanding of the phenomenon. For instance, if astronomical tides and storm surge peaks are coincident, storm tides can occur. A delay of few hours in one of these processes can lead to wrong predictions.**

*We agree with the reviewer's observation. Studies like this are fundamental for a better understanding of coastal processes in the Southwest Atlantic Ocean. Although our results show differences between the simulated output and in situ data, the values are close to the real ones, considering the complexity of the parameterization that modulates the ROMS and WRF models.*

**-Lines 552-562: This part of the manuscript looks more like the Introduction.**

*This text fragment presents other works that have similar previous results. Due to the absence of an effective oceanographic observation network using buoys, tide gauges, and drifters data, we do not have access to an optimum amount of in situ data. These data are essential for understanding the dynamic processes over these particular continental shelf waves and their characteristic parameters. Thus, we suggest that the observed and modeled free surface variations and wave propagation associated with storm-derived low-frequency values agree with other published works. For this reason, one of our authors suggested that we finalize the manuscript results by showing other works that observed similar atmosphere and oceanic conditions to ours. We believe that this paragraph is essential to conclude our ideas.*

*We agree with the reviewer and replace the fragment in introduction.*

**-Line 566 (Conclusions): For the reasons given above, I believe the conclusions are weakly proved. In essence, the conclusions are very superficial and do not contribute substantially to the knowledge of the process in the region.**

*We are grateful for the reviewer's suggestion. They were fundamental to reshape our manuscript. We edited our conclusions in order to improve our arguments and the manuscript quality.*